# Research on Mobile Network Parameters Using Unmanned Aerial Vehicles

**DOI:** 10.3390/s24175526

**Published:** 2024-08-26

**Authors:** Jan Warczek, Jarosław Kozuba, Marek Marcisz, Wiesław Pamuła, Krzysztof Dyl

**Affiliations:** Department of Air Transport, Faculty of Transport and Aviation Engineering, Silesian University of Technology, 44-100 Gliwice, Poland; jaroslaw.kozuba@polsl.pl (J.K.); marek.marcisz@polsl.pl (M.M.); wieslaw.pamula@polsl.pl (W.P.);

**Keywords:** unmanned aerial vehicles, mobile network field, mobile network parameters

## Abstract

The study of phenomena related to the propagation of electromagnetic waves is a necessity with the development of mobile telephony networks. This paper deals with the problem of the quality of mobile telephony signals. The study uses a BTS scanner, dedicated to scanning the frequency spectrum and decoding information from base stations. The device is capable of scanning mobile networks from the second, third, fourth, and fifth generation (2G/3G/4G/5G) using a passive sensor. The article presents preliminary results of monitoring cellular network signals using a scanner mounted on an unmanned aircraft. The collected data call for the need to evaluate the signal parameters of the cellular network depending on the altitude of signal reception. This property emerges as very important in the case of areas with a high level of urbanization.

## 1. Introduction

The study of phenomena related to the propagation of electromagnetic waves is common, but a clear increase in interest in this phenomenon is observed with the development of mobile telephony networks. The now classic GSM/GPRS (Global System for Mobile Communications with General Packet Radio Service) network solution is starting to fall behind the current needs related to mass information transfer. The current research focuses on the improvement of transmission efficiency. New-generation broadband mobile networks, e.g., UMTS/LTE (Universal Mobile Telecommunications System with Long-Term Evolution) are the result of research efforts. There are examples of applications of communication using a wireless network in various applications for which connection availability and reliability are crucial [1,2,3,4]. 

Today, the 5G standard with data transmission using millimeter waves is being introduced. This allows for increased data throughput yet limits the range of transmission in the presence of terrain obstacles. The next cycle of development of cellular network standards is 6G, which will mean not only a significant increase in data transmission speeds but also a change in the network architecture itself. It will cover three basic aspects: network infrastructure, communication technologies, and applications. In terms of communication, the 6G network will use microcells and macrocells with increased range; they will ensure uniform network coverage even in hard-to-reach areas. 

The 6G network will be based on a number of new technologies that will enable higher data transmission speeds and reduced latency. The use of technologies based on artificial intelligence (AI) methods is anticipated. AI can be used to optimize network management, dynamically allocate resources, and allow more efficient usage of resources. Thanks to AI, the 6G network will be able to adapt to changing network conditions and dynamically optimize the use of resources, which will contribute to improving the quality of services and enhancing energy efficiency. An important element of the 6G network will be the use of advanced antennas, such as phased array antennas. These antennas allow for more precise control of radio beams, which contributes to increased throughput and improved signal quality. In addition, MIMO (Multiple-Input Multiple-Output) antennas and beam-forming technology can also be used in the 6G network, which will increase the throughput and capacity of the network [5,6].

Currently, the monitoring of mobile network parameters is carried out as part of the inspection activities of telecommunication supervisory authorities. In Poland, it is the Polish Office of Electronic Communications that manages mobile operational teams in 16 delegations throughout Poland. The teams are equipped with specialized equipment and operate at distances of up to 150 km. Collecting network measurements is mostly performed manually at characteristic locations relative to the network base stations or at locations indicated as “dubious”. This approach is time consuming, demands significant human resources, and thus generates high operational costs. An alternative approach is the use of Unmanned Aerial Vehicles (UAVs) equipped with appropriate measuring devices. This enables the development of automatic monitoring systems collecting network parameter data not only at characteristic locations but also in the entire space enclosing the base stations of the networks. Automatic measurements can significantly reduce the data collection time and provide more detailed information on the properties of the cellular signals. The associated costs may be lower than in the case of manual measurements. 

The application of UAVs for measuring the parameters of wireless networks is presented in a number of papers. The authors of [7] propose tools for assessing the efficiency of wireless communication between base stations and UAVs. LTE parameters during flights are recorded and analyzed. GPS positioning data are used to determine distances and, combined with transmission parameters, enable the evaluation of TCP and UDP throughput in the space enclosing the base station. Work [8] presents the study of investigations of parameters of communication scenarios using LTE. A pair of UAVs is used for measuring key performance indicators (KPIs) at different flight altitudes. The paper presents the results of evaluating physical resource block (PRB) utilization, modulation, and coding scheme (MCS) class, throughput, and transmission power. The final conclusions confirm that controlling drones using LTE does not have a critical impact on network capacity. 

The results of wireless network studies in terms of availability and coverage quality carried out in the United States are presented in article [9]. A real-world assessment of the coverage and quality of the cellular network is necessary to assess the current reception properties of signals for network users. Simple, lightweight measuring devices, which can be mounted on a UAV, are proposed for signal quality evaluation. The solution is time efficient and provides long-range measurements. The successful application of UAVs for wireless network measurements is made possible by providing continuous situational awareness to the operator, who can control the drone from a long distance. This is particularly important when performing operations in various weather conditions [10]. 

One of the key features of using cellular networks for controlling UAVs is the capacity for two-way communication between the operator and the drone. The authors of [11] discuss the problems of cellular communication for drones flying at low altitudes. Data from field measurements collected during drone flights are transmitted using an LTE network in parallel with flight control signals. Simulation results are also presented to illustrate the performance of a network operating multiple drones simultaneously over a large area. The obtained results, analysis, and design insights help to better understand the application and performance issues associated with providing mobile connectivity for drones operating at low altitudes. Paper [12] presents the results of testing the quality of the LTE connection in communication with a drone. The study was conducted in South Korea. As altitude increases, the quality of the LTE signal deteriorates. However, the use of the LTE PUSCH Tx Power transmission protocol allows for the elimination of the effect of signal quality deterioration related to flight altitude.

The use of splines as a signal model can significantly improve the quality of signal processing due to the continuity of values and partial derivatives. This approach to the measurement of network parameters is presented in work [13]. The authors experimentally demonstrate that the use of splines allows for increased accuracy in determining network parameters. It is applied for monitoring the operation of services provided by the mobile operator, checking the readiness and availability of services, detecting network nodes through which quality deterioration occurs, collecting various quality metrics, and making comparisons with the base quality indicators.

In work [14], DroneSense is proposed, which is a system for measuring wireless signals in 3D space using autonomous drones. DroneSense reduces the time and effort required to collect measurements and is affordable and accessible to all users. It provides effective tools for quick analysis of wireless coverage and for testing wireless network propagation models. 

Another example of studying cellular network parameters is discussed in paper [15]. The problem of controlling UAV flight in the presence of traffic turbulence and noise, for monitoring surrounding network nodes, is addressed. Network topology prediction is performed using a general Kalman filtering-based framework with intermittent measurements for multiple targets. 

Cellular connectivity through ubiquitous cellular network coverage enables a variety of applications for long-range aerial inspections, surveillance, and drone monitoring. In these cases, the exchange of command and control data and sensor data such as video requires a reliable, low-latency communication link. Article [16] presents the results of studies on drone communication in an experimental 5G network. The results indicate that cellular communications generally maintain the reliability and latency of video and command and control data for long-haul flights, even when cellular network performance is optimized for ground-based devices.

Currently, Long Terminal Evolution (LTE) networks support broadband connectivity for users traveling at speeds up to 350 km/h, and support for speeds up to 500 km/h is being developed. Unfortunately, none of these efforts target air travelers due to a lack of antenna coverage. With 5G networks expected to keep everyone connected anywhere at any time, many operators are providing free onboard Wi-Fi via their own terrestrial networks or satellite connections. Unfortunately, both of these solutions have major drawbacks, with the latter providing very limited speeds and the former being expensive and unscalable. Article [17] describes the technical capabilities of expanding the existing LTE infrastructure for air-ground communication. The main challenges and obstacles in this path have been identified, such as uplink/downlink interference, frequent roaming, a large Doppler effect, and channel degradation.

To guarantee resource-efficient communication, an important research topic is to evaluate the channel quality in the device, including the channel quality for data transmission. In work [18], the correlation between the downlink channel quality indicators assessed in the device and the performance of the uplink system was analyzed. For this purpose, theoretical analysis and field measurements are performed in a dedicated LTE network as well as in the public LTE network. The results show that the reference received signal strength (RSRP) is a suitable indicator for very good uplink connectivity situations and that cell edge connectivity can be identified based on the received reference signal quality (RSRQ).

The development of wireless communication networks with increased capacity raises the problem of increasing their energy consumption. An attempt to remedy this is to use an independent power supply based on renewable energy sources. Paper [19] presents a model of the energy performance of an off-grid base location. The model consists of a solar energy system, a battery energy storage system, and a hydrogen energy storage system. It services various types of macrocells, microcells, picocells, or femtocells using broadband systems based on optical or microwave transmission. 

The goal of this study is to develop a quick and effective method, conforming with state regulations, for mapping cellular network parameters using unmanned aerial systems. Previous experience in the field of UAV applications in similar research areas indicates that the study results will allow us to obtain at least the same results that are currently achieved using traditional mobile equipment and yet carry out a much wider scope of mapping in both vertical and horizontal aspects. Previous tests also indicate that the interference of reflected, by obstacles, radio waves changes the propagation parameters and requires careful mapping. UAVs enable detailed 3D mapping and thus are suitable for performing network measurements. To a similar extent, flying drones are also used to collect measurement data regarding the area description of selected parameters, e.g., water in hard-to-reach places [20].

## 2. Research Object, Equipment, and Methods

The objective of this study was to obtain a spatial map of the parameters of the cellular network. The study is based on the use of a modern BTS scanner mounted on an unmanned aerial vehicle (UAV). The weight and dimensions of the scanner allow it to be mounted on a C3 class unmanned aircraft (<25 kg).

Mapping is performed using the Vespereye BTS Scout scanner (Vespereye, Poland) (Figure 1), dedicated to scanning and decoding information from base stations. It is a professional device for scanning 2G/3G/4G/5G mobile networks, allowing one to effectively detect the identity of BTS stations. The Scout device enables the determination of the availability of base stations in specific locations and confirms their ranges. The scanner sensor used in this research is completely passive. The scanner’s additional equipment is an independent GPS receiver. The device is characterized by high accuracy, a passive operating principle—using high-class radio technology, it operates in the temperature range of 0–60 °C—and a humidity range of 0–90%. The DJI Matrice 210 V2 UAV (Figure 1) was selected for the tests; its general technical specifications in terms of the carried payload are consistent with the parameters of the BTS scanner. 

Mapping was carried out in various areas differing in housing density. In the selected areas, the distribution of BTS stations was noted. The stations are equipped with masts differing in their construction and in the number of antennas placed on them. Figure 2 presents an example of a mast located in an area with low housing density. 

The range of mapping heights, at a test point, is limited by the UAV operating time on a set of batteries and the scanning time for obtaining information on BTS performance. A single flight allows for up to 8 mapping heights, and they were chosen to be 0, 2, 5, 10, 20, 30, 40, and 50 m. A vicinity with a 500 m radius around the mast was chosen for mapping the network parameters. 

In its final form, the mapping cycle consists of 128 measurements (16 locations × 8 heights). The measurement time at a single point is 12 min, which gives a total of 192 min of measurement.

Figure 3 presents the measurement plan. Three locations were omitted because of difficulties in entering the premises. There were 6 locations at a distance of 250 m from the mast and 7 locations 500 m from the mast. At each point, a series of 8 measurements were made at a fixed height. 

Figure 4 presents an instance of a measurement location with the UAV 2 m above the ground and a view of the surrounding housing, as shown in Figure 5. 

A comprehensive examination of the cellular field in accordance with official guidelines requires the development of an appropriate research procedure. Based on the adopted assumptions, a procedure scheme for examining a cellular network using a drone and a BTS scanner was prepared. The procedure is presented in Figure 6.

The scanner data collected during flights were placed in standard CSV files. The following network parameters were recorded: ARFCN—Absolute Radio Frequency Channel Number: All active devices listen to this channel and report to the BTS if necessary; this channel is located on one of the frequencies described by the channel frequency number;EARFCN—E-UTRA Absolute Radio Frequency Channel Number: All active devices listen to this channel and report to the BTS if necessary; this channel is located on one of the frequencies described by the channel frequency number;GPRS—confirmation of occurrence;LAC—unique BTS sector number within the location area code;shortCellId—shortened station identification number;longCellId—extended station identification number;MCC—country code;MNC—operator code;OP—network operator;RSCP—the power of the received signal in dBm, power of the signal correlated with a specific code sequence; this power is measured after the spectrum focusing operation;RSRP (Reference Signal Received Power)—the average power of the received pilot signals (reference signal) or the level of the received signal from the base station; the RSRP value is measured in dBm;RSRQ (Reference Signal Received Quality)—this parameter characterizes the quality of received pilot signals; the RSRQ value is measured in dB (dB);RxLev—level of RxLev signals (the minimum level for outdoor reception should be −97 dBm);shortCellId—a unique number used to identify each transmitting and receiving station (BTS);TAC—unique BTS sector number within the location area code;UARFCN—UTRA Absolute Radio Frequency Channel Number: All active devices listen to this channel and report to the BTS if necessary; this channel is located on one of the frequencies described by the channel frequency number.

RSCP and RSRQ values are used for describing the propagation properties of the cellular networks in the vicinity of the base station.

## 3. Results and Analysis

Table 1 presents the instances of collected scanner data in the 4G standard at location P9 at different flight heights. The data show that all Polish mobile operators use the mast for transmissions with different signal parameters.

Detected irregularities in the parameters of the 4G mobile network are marked in bold. RSRP (Reference Signal Received Power)—the average power of the received pilot signals (reference signal) or the level of the received signal from the base station. At RSRP = −120 dBm and below, the LTE connection may be unstable or not possible at all. RSRQ (Reference Signal Received Quality)—this parameter characterizes the quality of the received pilot signals. The RSRQ value is measured in dB (dB)—with an RSRQ above −10 dB, we can talk about a perfect connection. Values below −20 dB indicate very poor quality.

For the purpose of visualizing cellular field parameters, cross-sectional graphs of signal power and quality were proposed for the selected mobile network operator. Figure 7, Figure 8, Figure 9 and Figure 10 show the measurement results obtained at various distances from the base station. The results in Figure 7 and Figure 8 concern three network cells; Figure 9 and Figure 10 show three more network cells. The results show the vertical profile of signal strength and quality at selected measurement points located around the mobile phone mast.

As can be seen, as the distance from the BTS increases, the signal strength decreases, which is the expected effect. An interesting research issue is the complete loss of connection at point P13 located 500 m from the BTS. It should be mentioned that at this measurement point, the BTSs of a given operator located on base stations located much further away are revealed. As you can also notice, the signal quality at point P13 (especially at a low height above the surface) is quite poor—on the verge of smooth data transmission in the LTE standard. It should be assumed that a weak signal observed at a given measurement point is caused by a mismatch between the base station antenna power and the terrain and urban conditions. A network user in such a place will experience unexpected handovers between base stations, which significantly worsens the usefulness of the cellular network.

Based on the results of the cell field measurements, spatial signal power maps were developed. Examples of signal power distributions at different measurement heights obtained for EARFCN 1599 are shown in Figure 11 and Figure 12.

As can be observed, the RSRP values for the selected EARFCN 1599 as an example are quite irregular. The final form of the signal distribution is influenced by various factors: terrain, terrain obstacles, the presence of neighboring cell fields, and electromagnetic interference independent of the cell field.

## 4. Conclusions

The collected data allowed us to experimentally demonstrate the need to assess the parameters of the cellular network signal depending on height. The proposed solution using an unmanned aircraft as a carrier allows for testing network parameters in places inaccessible to classic methods based on cars. This is particularly important in the case of tests performed in areas with a high density of natural and artificial obstacles. Mapping the parameters of the mobile network allows for the assessment of its “quality” in area terms.

Further work is related to the development of a method for spatially imaging the signal quality of mobile networks in relation to terrain maps, taking into account height. This will allow, for example, to resolve issues related to problems with the availability of purchased services reported by mobile network users. 

## Figures and Tables

**Figure 1 sensors-24-05526-f001:**
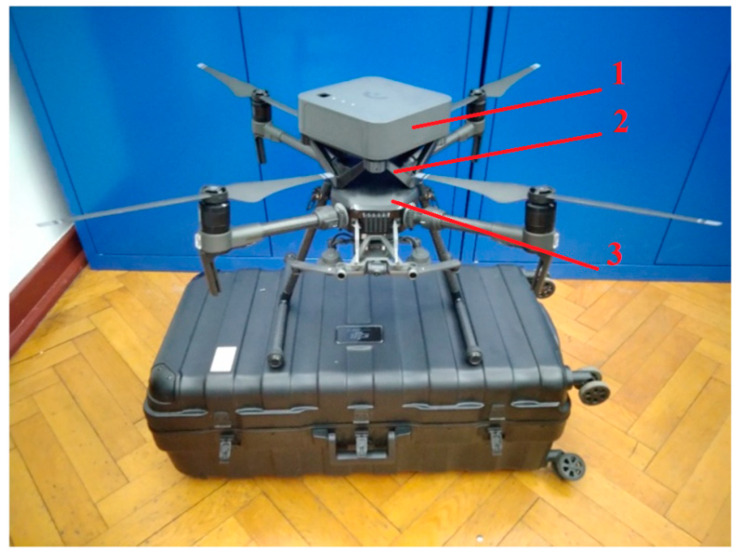
View of the measurement set used: 1—BTS scanner, 2—special mounting kit, and 3—DJI Matrice 210 V2 drone.

**Figure 2 sensors-24-05526-f002:**
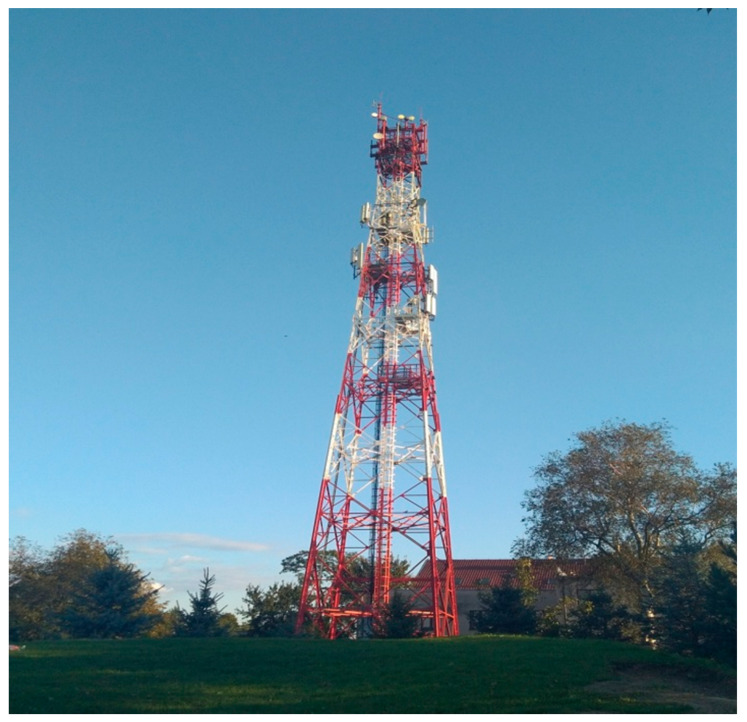
An example of a mast selected for mapping the mobile communication network parameters.

**Figure 3 sensors-24-05526-f003:**
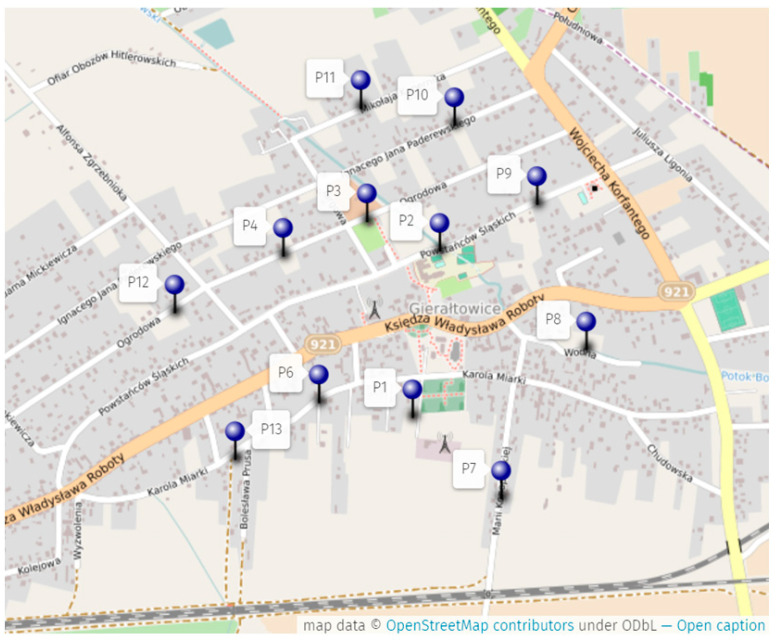
Measurement plan in the vicinity of a chosen BTS mast.

**Figure 4 sensors-24-05526-f004:**
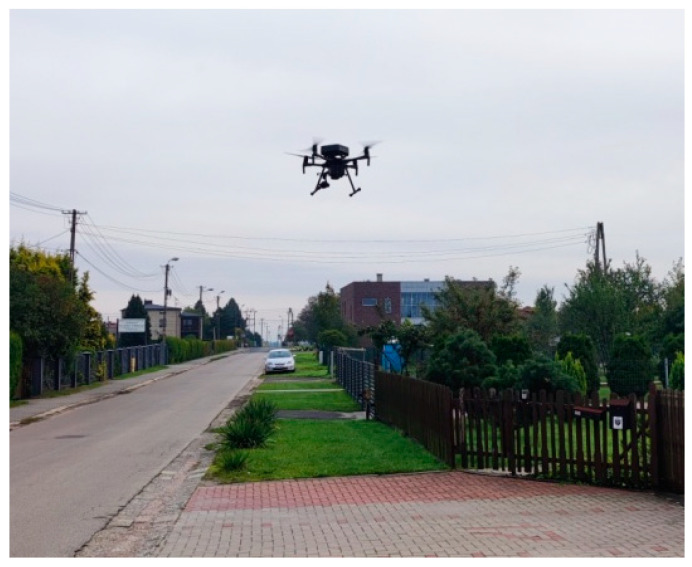
Measurement location with the UAV above ground.

**Figure 5 sensors-24-05526-f005:**
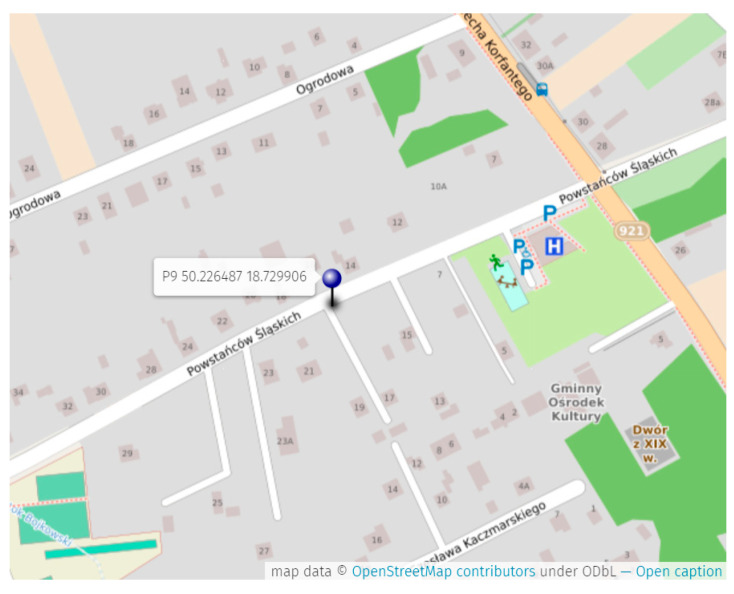
Position of point P9 with GPS data.

**Figure 6 sensors-24-05526-f006:**
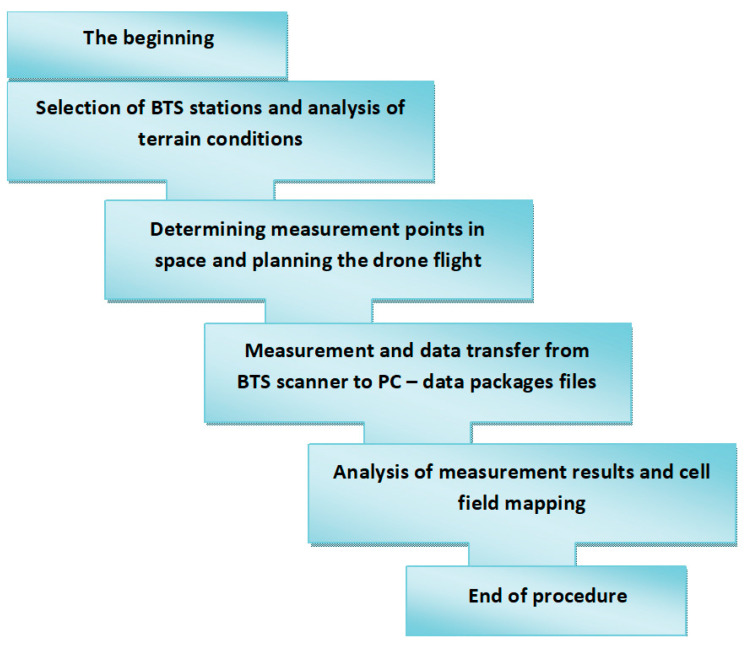
Diagram of the cellular network mapping procedure.

**Figure 7 sensors-24-05526-f007:**
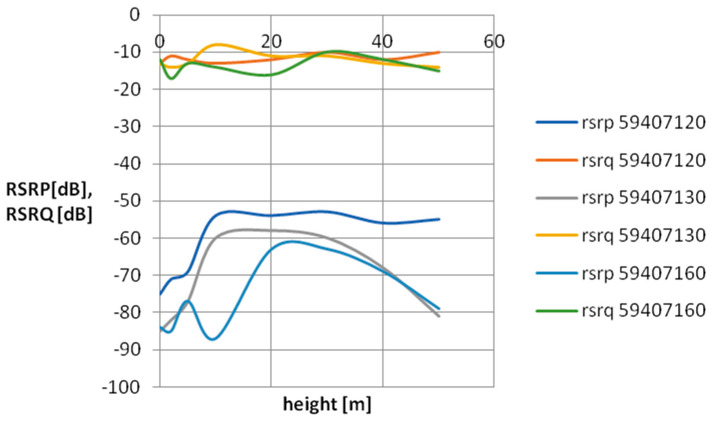
Vertical scanning results at point P2. The graph includes only parameters relating to the antennas of a specific network operator, characteristic for a given direction.

**Figure 8 sensors-24-05526-f008:**
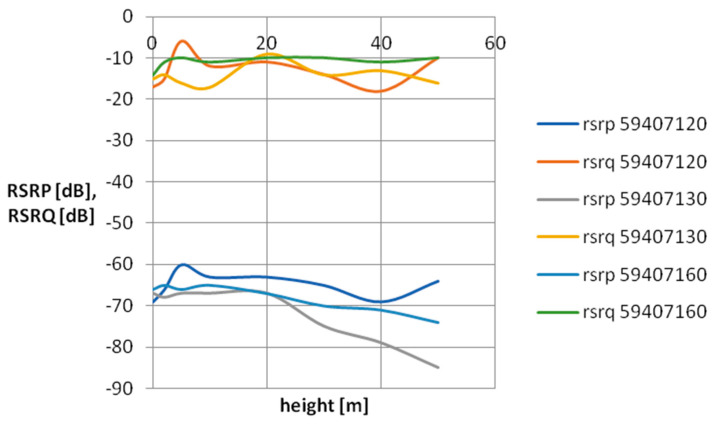
Vertical scanning results at point P9. The graph includes only parameters relating to the antennas of a specific network operator, characteristic for a given direction.

**Figure 9 sensors-24-05526-f009:**
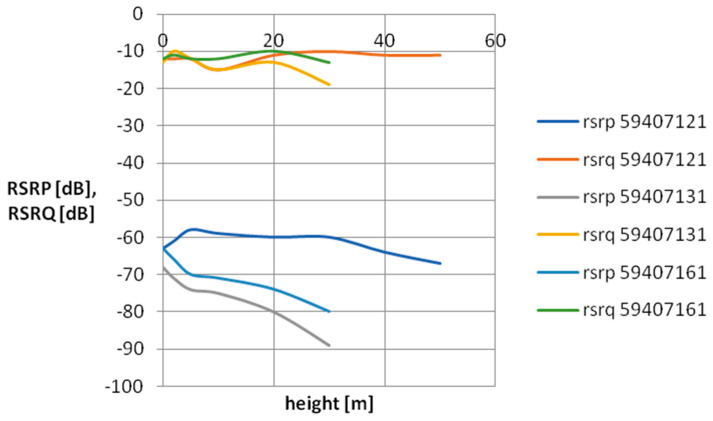
Vertical scanning results at point P6. The graph includes only parameters relating to the antennas of a specific network operator, characteristic for a given direction.

**Figure 10 sensors-24-05526-f010:**
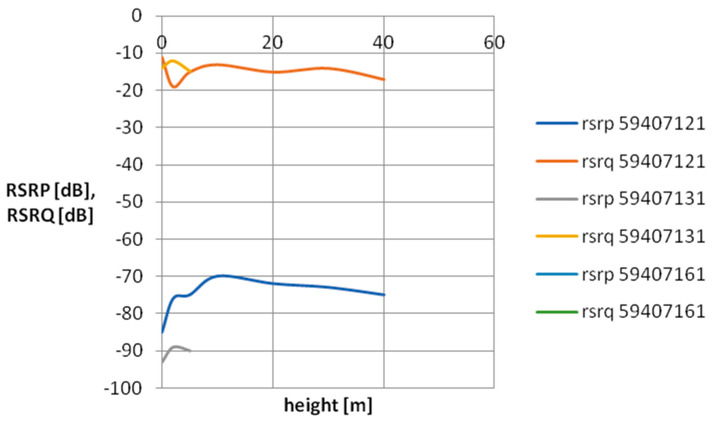
Vertical scanning results at point P13. The graph includes only parameters relating to the antennas of a specific network operator, characteristic for a given direction.

**Figure 11 sensors-24-05526-f011:**
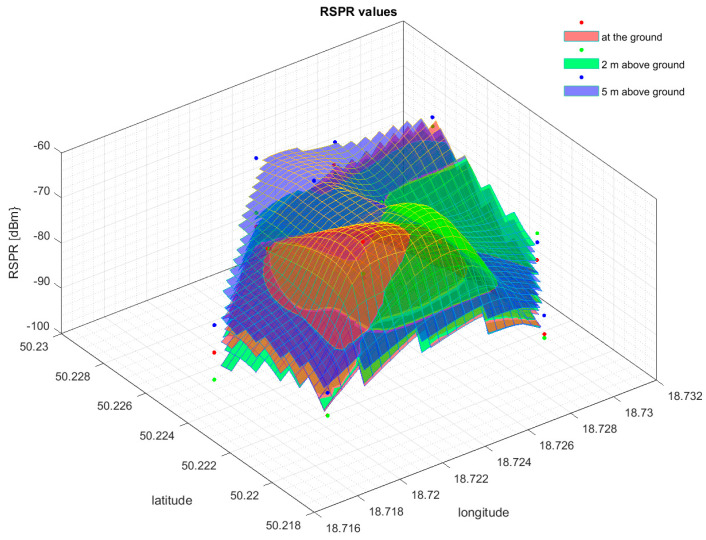
Reference signal received power distributions for EARFCN 1599 obtained at altitudes of 0, 2, and 5 m.

**Figure 12 sensors-24-05526-f012:**
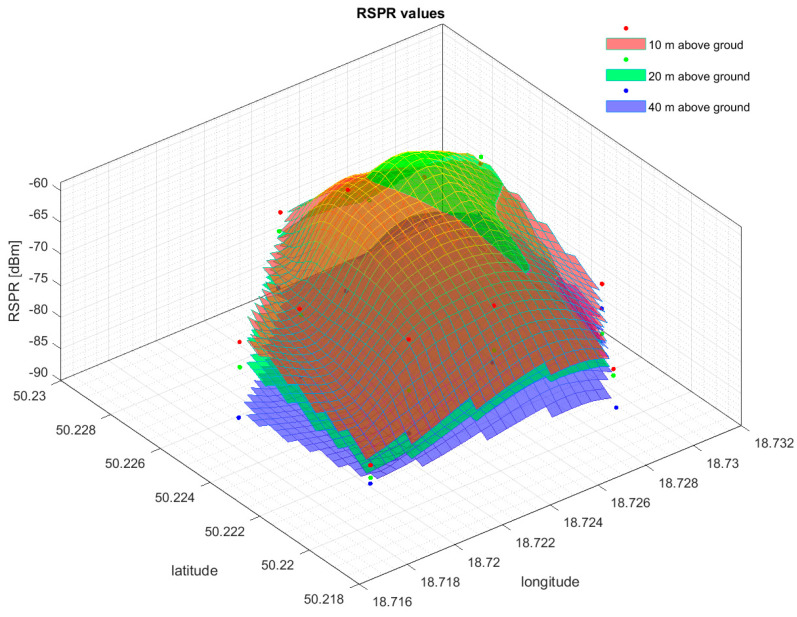
Reference Signal Received Power distributions for EARFCN 1599 obtained at altitudes of 10, 20, and 40 m.

**Table 1 sensors-24-05526-t001:** RSRP and RSRQ values at different heights above ground at location P9.

longCellId	EARFCN	RSRP [dBm]						RSRQ [dBm]					
		0	2 m	5 m	10 m	20 m	40 m	0	2 m	5 m	10 m	20 m	40 m
38A7B15	75	−66	−68	−66	−68	−67	−76	−13	−14	−10	−15	−9	−12
59407130	225	−67	−68	−67	−67	−67	−79	−15	−14	−16	−17	−9	−13
2548811	525	−86	−72	−77	−69	−72	−79	−13	−7	−**21**	−17	−18	−**24**
5355777	1300	−68	−64	−65	−66	−68	−66	−15	−12	−13	−16	−14	−12
2548810	1474	−80	−78	−81	−71	−66	−75	−17	−18	−17	−19	−8	−15
59407160	1599	−66	−65	−66	−65	−67	−71	−14	−11	−10	−11	−10	−11
38A7B33	1749	−66	−65	−67	−66	−69	−71	−13	−10	−12	−14	−10	−14
2548813	3350	−95	−82	−84	−73	−89	−92	−19	−7	−13	−11	−**27**	−**21**
5355797	3526	−72	−66	−65	−64	−64	−70	−14	−15	−15	−14	−12	−17
38A7B0B	6200	−68	−65	−61	−65	−64	−66	−11	−11	−12	−10	−14	−13
2548812	6275	−73	−68	−72	−63	−69	−68	−12	−17	−10	−17	−16	−17
59407120	6350	−69	−66	−60	−63	−63	−69	−17	−15	−6	−12	−11	−18

## Data Availability

The raw data supporting the conclusions of this article will be made available by the authors on request.

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
