# Peer review of "Research on Mobile Network Parameters Using Unmanned Aerial Vehicles"

_sensors, 2024, doi:10.3390/s24175526_

Round 1

Reviewer 1 Report

Comments and Suggestions for Authors

By using the Vespereye BTS scout scanner, this paper provides preliminary results of testing cellular UAV network. In my opinion, the contributions of this paper are not enough. My comments are provided as follows:

1. In this paper, the authors discuss the GSM/GPRS, UMTS/LTE techniques, which are outdated technologies. The practicality of this work should be improved. Advanced 5G/6G technologies should be discussed.

2. The presentation should be improved, such as Paper (1) …, Article (2) …, Paper (3) …

3. The introduction section is divided into too many paragraphs, which should be improved to facilitate the read and understanding of the readers.

4. The derived parameters and data may be useful for engineering application, however, the innovation and novelty of this work are not enough.

5. Some figures are too large, such as Fig. 4.

6. In addition to RSRP and RSRQ, more in-depth results should be provided.

Author Response

Dear Reviewer, Thank you very much for your advice and questions. Below are our answers.

  1. In this paper, the authors discuss the GSM/GPRS, UMTS/LTE techniques, which are outdated technologies. The practicality of this work should be improved. Advanced 5G/6G technologies should be discussed.

Response:

Thank you for this comment. We have added information about 5G and 6G in the introduction.

  1. The presentation should be improved, such as Paper (1) …, Article (2) …, Paper (3) …

Response:

The description of research by other authors has been improved. 

  1. The introduction section is divided into too many paragraphs, which should be improved to facilitate the read and understanding of the readers.

Response:

Thanks for the tip - the way content is divided has been changed. 

  1. The derived parameters and data may be useful for engineering application, however, the innovation and novelty of this work are not enough.

Response:

The obtained results allow for spatial analysis of cellular network parameters. The study of cellular network parameters is currently limited by legal requirements only to studies at points located on the ground surface. An additional limitation of currently used methods is limited field access resulting from property rights. Using a drone as a carrier of a cellular network scanner solves these problems.

  1. Some figures are too large, such as Fig. 4.

Response:

 Thank you for your comments - the size of figures has been changed.

  1. In addition to RSRP and RSRQ, more in-depth results should be provided.

Response:

 The cellular network scanner used is a device compliant with the requirements of the Office of Electronic Communications and its design cannot be tampered with in an unauthorized manner. As a novelty in our work, spatial distributions for RSRP were added (Fig. 9 and 10).

Changes to the article content are marked in red. Thank you again for reviewing our article.

Reviewer 2 Report

Comments and Suggestions for Authors

This paper presents preliminary results of testing cellular network signals using a scanner mounted on an unmanned aircraft. Some comments are shown as follows:

1. The literature research section of this paper is written loosely and lacks a high degree of concision, which is not friendly to the current situation research of scientific papers, making it difficult for readers to systematically grasp and clarify the relevant research. It is necessary to summarize and analyze existing research, rather than displaying the research content of each paper.

2. The innovation of this paper is not clear. It only uses drones as a substitute in the solution based on automotive measurement, and its contribution is not high.

3. The literature of this paper is relatively old and needs to be updated with the latest relevant research.

4. In the simulation section, the impact of hardware devices such as scanner functions and antennas on signal acquisition effectiveness needs to be analyzed.

Comments on the Quality of English Language

Some sentences are expressed too long, which is not conducive to reading. Please check and modify.

Author Response

Dear Reviewer, thank you very much for your review of our work. Below you will find our responses.

  1. The literature research section of this paper is written loosely and lacks a high degree of concision, which is not friendly to the current situation research of scientific papers, making it difficult for readers to systematically grasp and clarify the relevant research. It is necessary to summarize and analyze existing research, rather than displaying the research content of each paper.

Response:

Thank you for your comment and we agree with it. The literature review was improved and the scope of own research was clarified.

  1. The innovation of this paper is not clear. It only uses drones as a substitute in the solution based on automotive measurement, and its contribution is not high.

Response:

Thank you for the tip. The advisability of using drones in cell field research has been clarified.

  1. The literature of this paper is relatively old and needs to be updated with the latest relevant research.

Response:

 Newer literature items have been added.

  1. In the simulation section, the impact of hardware devices such as scanner functions and antennas on signal acquisition effectiveness needs to be analyzed.

Response:

This is a very valuable comment. Due to cooperation with the Office of Electronic Communications, the scanner used meets the requirements specifying the parameters of devices for testing cellular networks. In the future, we plan to conduct extended tests of cellular networks.

Changes to the article content are marked in red.

 Thank you again for reviewing our article.

Reviewer 3 Report

Comments and Suggestions for Authors

Research of Mobile Networks Parameters Using UAVs (MDPI Sensors Journal Review)

July 2024

Thank you for submitting your work to the MDPI Sensors Journal. 1 Paper summary.

The research paper explores the impact of altitude on mobile network signal quality using UAVs equipped with the Vespereye BTS Scout scanner, a device capable of scanning and identifying base stations across 2G to 5G networks. This study focuses on assessing mobile telephony signals in urbanized areas, showing that altitude plays a significant role in network performance, backed by data collected from the UAV-mounted scanner. The primary suggestion would be to improve the conciseness and compactness of the paper to create more impact for the readers. Suggestions: 1. The citation style needs to be verified and fixed. It would be better to cite multiple papers on a similar topic instead of adding a paragraph for each new paper cited. Additionally, it’s essential to include what you learned and how it applies to your paper from the cited work. The language of the review could be improved, but the critical analysis is missing. Making these changes will make the work more concise and impactful. Emphasize what new insights or methodologies your UAV approach brings to mobile network research, mainly how it differs from or improves upon existing methods. 2. The detailed specifications of the DJI Matrice 210 V2 UAV are typically accessible directly from the manufacturer; therefore, they should only be included if they have been customized specifically for this research. 3. It would be beneficial to discuss any regulatory or logistical challenges faced during the study, such as obtaining permission to fly UAVs near cellular infrastructure and how these were overcome. 4. Additionally, consider consolidating the tabular data on network assessments into fewer tables to streamline the content and enhance readability. 5. Please also verify the caption for Figure 5 and fix similar errors. 6. More detailed explanations about the methodology for setting up the measurements, including any equipment calibration and the exact data collection process, would enhance the study’s reproducibility. 7. Incorporate more advanced statistical analysis to evaluate the significance of the results across different altitudes and points. This might include regression or variance analysis to understand height’s impact on signal quality better. 8. Expand on the analysis of complete signal loss at specific points, such as point P13. Discuss potential causes and implications in more detail to provide a more comprehensive understanding of network vulnerabilities. 9. Include comparisons with other studies or benchmarks in the field to contextualize the findings and validate the methods used.

The paper has some merit and can be published with suggested improvements. 

Comments on the Quality of English Language

Fix the minor errors mentioned and proofread.

Author Response

Thank you very much for reviewing our article. We are posting answers to your suggestions:

  1. The citation style needs to be verified and fixed. It would be better to cite multiple papers on a similar topic instead of adding a paragraph for each new paper cited. Additionally, it’s essential to include what you learned and how it applies to your paper from the cited work. The language of the review could be improved, but the critical analysis is missing. Making these changes will make the work more concise and impactful. Emphasize what new insights or methodologies your UAV approach brings to mobile network research, mainly how it differs from or improves upon existing methods.

Response:

Thank you for this suggestion. We tried to select articles that best match the subject of our work. The literature review has been revised and expanded.

  1. The detailed specifications of the DJI Matrice 210 V2 UAV are typically accessible directly from the manufacturer; therefore, they should only be included if they have been customized specifically for this research.

Response:

Thank you for this suggestion. The specification of the drone used has been shortened to the essential data.

  1. It would be beneficial to discuss any regulatory or logistical challenges faced during the study, such as obtaining permission to fly UAVs near cellular infrastructure and how these were overcome.

Response:

Research conducted in accordance with applicable Polish regulations regarding drone flights. Any application of this method in other countries must be correlated with an analysis of applicable legal regulations.

  1. Additionally, consider consolidating the tabular data on network assessments into fewer tables to streamline the content and enhance readability.

Response:

This is a valuable suggestion, thank you. Limited the number of results in tables to only 4G networks.

  1. Please also verify the caption for Figure 5 and fix similar errors.

Response:

We are very sorry for the incorrect descriptions of figures. Descriptions under figures have been corrected.

  1. More detailed explanations about the methodology for setting up the measurements, including any equipment calibration and the exact data collection process, would enhance the study’s reproducibility.

Response:

 The scanner used complies with the requirements specifying the parameters of devices for testing cellular networks. The device has an appropriate certificate in accordance with Polish law.

  1. Incorporate more advanced statistical analysis to evaluate the significance of the results across different altitudes and points. This might include regression or variance analysis to understand height’s impact on signal quality better.

Response:

 The results obtained with the cellular network scanner already include statistical analysis. The values ​​provided by the scanner are in accordance with the guidelines for cellular network testing. New figures were added to the article to improve the visualization of the effect of altitude on cellular network parameters.

  1. Expand on the analysis of complete signal loss at specific points, such as point P13. Discuss potential causes and implications in more detail to provide a more comprehensive understanding of network vulnerabilities.

Response:

Thanks for the tip. Additional description of the obtained results was added.

  1. Include comparisons with other studies or benchmarks in the field to contextualize the findings and validate the methods used.

Response:

Based on the literature review, no similar studies were identified.

Changes to the article content are marked in red.

Thank you again for reviewing our article.

Round 2

Reviewer 1 Report

Comments and Suggestions for Authors

The authors have replied to my previous comments. I still have some comments:

1. The references must be cited in order. Moreover, "Authors in (1)" should be "Authors in [1]", "Work (2)" should be "Work [2]", "article (3)" should be "article [3]", et al.

2. "The table 1 present ..." should be changed to "Table 1 presents ...".

3. Please provide high-resolution images for Figs. 3 and 4. 

Author Response

Thank you again for reviewing our article. We have provided responses to the comments in the review below.

Comments and Suggestions for Authors

The authors have replied to my previous comments. I still have some comments:

  1. The references must be cited in order. Moreover, "Authors in (1)" should be "Authors in [1]", "Work (2)" should be "Work [2]", "article (3)" should be "article [3]", et al.

Response:

Thank you for your comment – ​​the citation has been corrected.

  1. "The table 1 present ..." should be changed to "Table 1 presents ...".

Response:

We apologize for this error – it has been corrected.

  1. Please provide high-resolution images for Figs. 3 and 4. 

Response:

Figures 3 and 4 have been changed to be more readable.

Thank you again for your insightful and substantive review of our work. Changes to the article content are marked in red.

Reviewer 2 Report

Comments and Suggestions for Authors

I have no other comments.

Author Response

Comments and Suggestions for Authors

I have no other comments.

Response:

Thank you for your positive review of our article.

Reviewer 3 Report

Comments and Suggestions for Authors

Comments for V2:

Thank you for making the changes in the revised document. The document has some merit in novelty; however, the authors have overlooked some critical aspects that must be addressed before it can be published in this journal.

According to the MDPI Sensors Journal \textit{Instructions for Authors}, the recommended length for an Article submission is 16 or more pages. The current submission is approximately 12 pages, categorizing it as a Communication. The authors must ensure compliance with the submission guidelines, which can be found at: (https://www.mdpi.com/journal/sensors/instructions).

The journal title is 'Sensors,' yet the terms 'sensors' and 'sensing' are conspicuously absent from the document. Considering the submission is classified under the 'Section: Communications' and 'Special Issue: Advancements and Applications of UAV Communications with RF, Microwave, and mmWave Techniques,' it is crucial to incorporate relevant terminology. This aligns with the journal's thematic focus and enhances the visibility and relevance of your submission within the field.

It is good practice to annotate figures and label the essential specifications, especially for Figure 1.

It would be helpful to readers if the authors could add a block diagram/ flowchart explaining the system functionality and methodology of the experiments. More details about how the experiment is performed could be added. 

Are there any specific methods of flying the UAV in a pattern or type of trajectory matter in the sensing results?

Verify the formatting of the reference's bibliography in the PDF. For example, [7] has the doi and page numbers without space. Also, there is no section 4 (after section 3, section 5 is added, skipping 4).

It could be beneficial for the readers to include a discussion and future work section that outlines the results obtained in section 3. The current section on future work needs to be more specific and aligned with the results presented. It mentions further developments but does not detail particular strategies or next steps based on the study's outcomes, missing an opportunity to propose a clear direction for subsequent research.

The abstract states, `The paper deals with the problem of the quality of mobile telephony signals.' This statement is vague and does not specify the unique challenges the study addresses. A more detailed problem statement identifying specific issues with signal quality in urbanized environments would strengthen the introduction.

The use of UAVs for data collection is mentioned: `The scanner data collected during flights is placed in standard CSV files.' However, the data validation process or how the UAVs were configured to ensure accurate data collection needs to be explained. Detailing these aspects would enhance the methodological rigour of the study.

`Using splines as a signal model can significantly improve signal processing quality.' While the paper mentions advanced analytical techniques, it lacks a comparative analysis with other data smoothing or predictive models. Discussing why splines were chosen over other potential models would clarify this choice.

Comments on the Quality of English Language

Please verify the English language using appropriate tools or get it professionally checked. There are instances of minor grammatical errors that could be improved.  For instance, the line 'The table 1 present the instances of collected scanner data in 4G standard at location P9 at different flight heights. ' needs to be: 'Table 1 presents the instances of collected scanner data in 4G standard at location P9 at different flight heights.  '

Author Response

Once again, thank you for your insightful and substantive review of our work. We hope that our responses will meet all expectations and allow for the acceptance of our work. Below are the responses to the comments included in the review.

Comments and Suggestions for Authors

Comments for V2:

 According to the MDPI Sensors Journal \textit{Instructions for Authors}, the recommended length for an Article submission is 16 or more pages. The current submission is approximately 12 pages, categorizing it as a Communication. The authors must ensure compliance with the submission guidelines, which can be found at: (https://www.mdpi.com/journal/sensors/instructions).

Response:

Thank you for your comment. The article has been expanded as suggested.

The journal title is 'Sensors,' yet the terms 'sensors' and 'sensing' are conspicuously absent from the document. Considering the submission is classified under the 'Section: Communications' and 'Special Issue: Advancements and Applications of UAV Communications with RF, Microwave, and mmWave Techniques,' it is crucial to incorporate relevant terminology. This aligns with the journal's thematic focus and enhances the visibility and relevance of your submission within the field.

Response:

 Thank you for your comment. The article content has been changed, with particular attention to the specifics of the journal.

It is good practice to annotate figures and label the essential specifications, especially for Figure 1.

Response:

Figure 1 has been changed – a description of the elements of the measurement set has been added. In accordance with the comments from the first stage of the review, the specifications provided by the manufacturers of the devices used to build the measurement set have been removed from the article. In our opinion, the design of the base for mounting the scanner does not contribute anything significant to the content of the article.

It would be helpful to readers if the authors could add a block diagram/ flowchart explaining the system functionality and methodology of the experiments. More details about how the experiment is performed could be added.

Response:

Thank you for this valuable comment. The relevant flowchart of the measurement procedure has been added to the article.

Are there any specific methods of flying the UAV in a pattern or type of trajectory matter in the sensing results?

Response:

To our knowledge, there are no specific models for performing a flight during a cellular network survey. The authors have proposed their own range of measurements performed in flight based on preliminary studies on the applied elements of the measurement set.

Verify the formatting of the reference's bibliography in the PDF. For example, [7] has the doi and page numbers without space. Also, there is no section 4 (after section 3, section 5 is added, skipping 4).

Response:

Thank you for your thorough review and for detecting our errors. The bibliography and chapter numbering have been corrected.

It could be beneficial for the readers to include a discussion and future work section that outlines the results obtained in section 3. The current section on future work needs to be more specific and aligned with the results presented. It mentions further developments but does not detail particular strategies or next steps based on the study's outcomes, missing an opportunity to propose a clear direction for subsequent research.

Response:

Thank you for this comment. At the moment we do not have ready solutions and therefore the proposal for future work has been described somewhat generally. We hope that in the next publications we will be able to present, first of all, a detailed procedure for mapping the cellular network and determining the identification of coverage problems.

The abstract states, `The paper deals with the problem of the quality of mobile telephony signals.' This statement is vague and does not specify the unique challenges the study addresses. A more detailed problem statement identifying specific issues with signal quality in urbanized environments would strengthen the introduction.

Response:

We agree with this comment. Nevertheless, it is difficult to clearly determine the quality of a mobile network. Our research is conducted in cooperation with the Office of Electronic Communications, which often faces the problem of resolving disputes between the network operator and its users. For this reason, a verification tool is needed to confirm the reports of mobile network users about its incorrect operation. This type of tool is a further goal of our research work.

The use of UAVs for data collection is mentioned: `The scanner data collected during flights is placed in standard CSV files.' However, the data validation process or how the UAVs were configured to ensure accurate data collection needs to be explained. Detailing these aspects would enhance the methodological rigour of the study.

Response:

Thank you for this comment. The BTS scanner used in our research is a device independent of the carrier used (drone). The further goal of the work will be to build a flying device with an integrated scanner. The currently proposed data file format is a working version and we assume the possibility of changing to another method of collecting results.

`Using splines as a signal model can significantly improve signal processing quality.' While the paper mentions advanced analytical techniques, it lacks a comparative analysis with other data smoothing or predictive models. Discussing why splines were chosen over other potential models would clarify this choice.

Response:

This commentary concerns the description of the studies cited by other Authors. Due to the necessity of limiting the literature review, not all the threads discussed in the cited work were cited in our work.

Comments on the Quality of English Language

Please verify the English language using appropriate tools or get it professionally checked. There are instances of minor grammatical errors that could be improved.  For instance, the line 'The table 1 present the instances of collected scanner data in 4G standard at location P9 at different flight heights. ' needs to be: 'Table 1 presents the instances of collected scanner data in 4G standard at location P9 at different flight heights.  '

Thank you for your comments – the article text has been corrected. Changes to the article content are marked in red.

Round 3

Reviewer 3 Report

Comments and Suggestions for Authors

Thank you for making the changes and for all the effort and contribution from your team. The paper is accepted in its current form.